# Gut dysbiosis contributes to the development of Budd-Chiari syndrome through immune imbalance

Qinwei Lu,[1,2,3] Rongtao Zhu,[1,2,3] Lin Zhou,[2,3,4] Ruifang Zhang,[5] Zhen Li,[6] Peng Xu,[7] Zhiwei Wang,[7] Gang Wu,[8] Jianzhuang Ren,[8] Dechao Jiao,[8] Yan Song,[8] Jian Li,[1,2,3] Weijie Wang,[1,2,3] Ruopeng Liang,[1,2,3] Xiuxian Ma,[1,2,3] Yuling Sun[1,2,3]

**ABSTRACT** Budd-Chiari syndrome (B-CS) is a rare and lethal condition characterized by hepatic venous outflow tract blockage. Gut microbiota has been linked to numerous hepatic disorders, but its significance in B-CS pathogenesis is uncertain. First, we performed a case-control study ($N_{case}$ = 140, $N_{control}$ = 63) to compare the fecal microbiota of B-CS and healthy individuals by metagenomics sequencing. B-CS patients' gut microbial composition and activity changed significantly, with a different metagenomic makeup, increased potentially pathogenic bacteria, including *Prevotella*, and disease-linked microbial function. Imbalanced cytokines in patients were demonstrated to be associated with gut dysbiosis, which led us to suspect that B-CS is associated with gut microbiota and immune dysregulation. Next, 16S ribosomal DNA sequencing on fecal microbiota transplantation (FMT) mice models examined the link between gut dysbiosis and B-CS. FMT models showed damaged liver tissues, posterior inferior vena cava, and increased *Prevotella* in the disturbed gut microbiota of FMT mice. Notably, B-CS-FMT impaired the morphological structure of colonic tissues and increased intestinal permeability. Furthermore, a significant increase of the same cytokines (IL-5, IL-6, IL-9, IL-10, IL-17A, IL-17F, and IL-13) and endotoxin levels in B-CS-FMT mice were observed. Our study suggested that gut microbial dysbiosis may cause B-CS through immunological dysregulation.

**IMPORTANCE** This study revealed that gut microbial dysbiosis may cause Budd-Chiari syndrome (B-CS). Gut dysbiosis enhanced intestinal permeability, and toxic metabolites and imbalanced cytokines activated the immune system. Consequently, the escalation of causative factors led to their concentration in the portal vein, thereby compromising both the liver parenchyma and outflow tract. Therefore, we proposed that gut microbial dysbiosis induced immune imbalance by chronic systemic inflammation, which contributed to the B-CS development. Furthermore, *Prevotella* may mediate inflammation development and immune imbalance, showing potential in B-CS pathogenesis.

**KEYWORDS** Budd-Chiari syndrome, gut microbiota, gut microbiota dysbiosis, fecal microbiota transplantation, immune imbalance

B udd-Chiari syndrome (B-CS) is a rare congestive hepatopathy caused by an obstruction of the hepatic venous outflow tract without cardiac or pericardial illness, resulting in diminished liver outflow, gastrointestinal bleeding, fever, and hypoxic liver injury (1, 2). The obstruction causing B-CS is usually located in the small or large hepatic veins or the suprahepatic portion of the inferior vena cava (IVC) (3). However, the difference in B-CS characteristics between West and Asia should be recognized (4). Increased blood viscosity is the main pathophysiological mechanism of B-CS in Western countries (4, 5). Among them, myeloproliferative disorders, pregnancy, antiphospholipid syndrome, use of coagulants, and antithrombin III protein C and S deficiency are all

Address correspondence to Yuling Sun, ylsun@zzu.edu.cn.

Qinwei Lu and Rongtao Zhu contributed equally to this article. Author order was determined both alphabetically and in order of increasing seniority.

The authors declare no conflict of interest.

See the funding table on p. 16.

important risk factors for B-CS (6–8). Inversely, membranous webs have long been considered the leading cause of B-CS in Eastern countries (4, 9). The pathogenesis of this morbidity is also vague to date (1). Emerging studies have indicated that B-CS should be considered a liver disease closely associated with gut microbiota dysbiosis and immune imbalance (10, 11).

Increasing data suggest that the gut microbial population has a role in metabolism, intestinal barrier function, intestinal permeability, and immune system maturation (12, 13). The intestinal barrier restricts the translocation of intestinal pathogenic bacteria and their toxins to tissues and organs beyond the intestinal cavity, preventing the host from being invaded by endogenous microorganisms and toxins (14). Gut-inhabiting beneficial bacteria could prevent the colonization of pathogens by regulating the intestinal acidic environment, producing antimicrobial peptides, and competing for adhesion sites and nutrition, which was regarded as a natural barrier against pathogenic bacteria infection (15, 16). Although these microorganisms inhabit the intestine, they may cause systemic effects (17). Numerous investigations showed that gut microbial dysbiosis impaired intestinal processes and damaged nearby and distant organ systems like the liver and brain (18, 19). Importantly, gut microbiota has also been shown to be the central or contributing factor of multiple diseases, including diarrhea, obesity, diabetes, and non-alcoholic fatty liver (20, 21). However, gut microbial homeostasis is easily influenced by multiple factors like external environment, host physiology, and nutritional and health status (22, 23).

The liver and gut interact bidirectionally, causing hepatic illnesses like cirrhosis, inflammation, and hepatocellular carcinoma (24). Intestinal bacteria and their derivatives and metabolites can be transferred to the liver by the portal vein, causing liver-related disease deterioration during gut microbial dysbiosis (25). However, the gut microbiota and B-CS interaction is unclear. Our previous investigations showed that B-CS patients have a disturbed gut microbiota composition (10), but there is a lack of causative analysis. In this study, we investigated the detailed gut microbiota profile to explore its crucial role in triggering B-CS-like phenotypes by fecal microbiota transplantation (FMT). Moreover, the potential mechanistic relationship between gut microbiota and B-CS development was explored for the first time.

## MATERIALS AND METHODS

### Study subjects

B-CS patients and their healthy peers were recruited at the First Affiliated Hospital of Zhengzhou University. During this period, 140 patients with B-CS and 63 healthy members were included. According to the present practice guidance by the American Association for the Study of Liver Diseases (1), patients in the cohort were diagnosed using a multidetector computed tomography three-dimensional vascular reconstruction technique combined with color Doppler ultrasound. The patients and healthy controls had not taken probiotics or dietary yogurt for 1 month and had not taken antibiotics in 3 months. None of the subjects suffered from any disease affecting gut microbiota, including obesity, hypertension, diabetes, intestinal diseases, and constipation. Meanwhile, other liver diseases were also excluded, like liver tumors and biliary stones. All participants provided written informed consent for participation in the study. The study protocol was approved by the local ethics committee.

### Experimental design and treatment

FMT was administered using fresh stools from B-CS patients and healthy donors. Then, feces were immediately poured into sterile saline and resuspended, and the dissolved feces were centrifuged for enough suspension. Six-week-old healthy C57BL/6J mice ($n$ = 20) with similar weight were bought from Shanghai Cell and Molecular Biology Research Center (Shanghai, China). The mice we selected were raised in standard-size

cages under the recommended illumination time (12-h light/dark cycle) and in a clean environment for 42 days. All mice were fed sterile food and water *ad libitum* in this experiment. All mice were acclimatized to the habitat for 3 days to reduce stress from unexpected environmental changes. Subsequently, all the mice were randomly divided into two groups, with 10 mice in each group: a control group and fecal microbiota transplantation from patients with the Budd-Chiari syndrome group (B-CS-FMT group). The B-CS-FMT mice were forced to consume 200 µL of fecal suspension daily for 42 days, while the control mice received the same amount of suspension from healthy controls. The clinical symptoms of mice were recorded periodically during the experiment, and death, anorexia, and depression were regarded as abnormal.

## Sample acquisition

For B-CS patients and healthy controls, fresh stool samples were obtained and stored at −80°C before the shotgun metagenomics sequencing, and blood samples were taken and spun into serum samples for future testing. Before the mice were slaughtered, fresh stool samples were collected for 16S rDNA amplicon sequencing, and blood samples were collected for cytokine analysis. Then, all the mice were euthanized. The intestine and liver of each euthanized mouse were separated from the abdominal cavity, and different intestinal segments were knotted using cotton ropes to minimize cross-contamination. Mid-colonic segments were promptly collected, snap-frozen in liquid nitrogen, and stored at −80°C until DNA extraction. Colon and liver samples were fixed in 4% paraformaldehyde overnight. Subsequently, the hematoxylin and eosin (H&E) staining was conducted as previously described (26).

## Immunofluorescence staining

After 24 h in 4% paraformaldehyde, the liver and colon were embedded in paraffin. The histological investigation used H&E-stained serial sections. To prevent nonspecific adsorption, untreated samples from the same batch were incubated with goat serum (1:10 in PBS) at 37°C for 1 h. Subsequently, specific antibodies (first antibody, 1:200 in PBS) were added to the sample surfaces and incubated at 37°C for 30 min. After washing thrice with PBS, FITC-conjugated goat anti-mouse IgG antibody (second antibody, 1:100 in PBS) was added to the samples and incubated at 37°C for 30 min. The primary antibodies used to observe the liver and vessels were rabbit monoclonal anti-mouse CD31, CS34, vWF, and ICAM antibodies, whereas the antibodies used to observe the colon were rabbit monoclonal anti-mouse Z0-1 and Occludin antibody. Finally, the samples were thoroughly rinsed thrice with PBS and monitored by fluorescence microscopy (Leica, Germany).

## Immunohistochemical staining

Colon paraffin slices were immunohistochemically stained to determine Occludin and Zonula Occludens-1 (ZO-1) subcellular localization. As per previous investigations, immunohistochemistry staining was done (26).

## Serum cytokine and endotoxin analysis

All blood samples were placed statically for 1 h and then centrifuged at 3,500 rpm for 10 min to obtain serum. The serum cytokine of human (IL-8, IP-10, Eotaxin, TARC, MCP-1, RANTES, MIG, ENA-78, MIP-3α, GROα, I-TAC, MIP-1β, TSLP, IL-1α, IL-1β, GM-CSF, IFN-α2, IL-23, IL-12p40, IL-12p70, IL-15, IL-18, IL-11, IL-27, IL-33, IL-5, IL-13, IL-2, IL-6, IL-9, IL-10, IFN-γ, TNF-α, IL-17A, IL-17F, IL-4, IL-22, and IL-21) and mice (IL-8, I-TAC, MIP-1α, TARC, IP-10, TSLP, IL-1α, IL-1β, GM-CSF, IFN-α2, IL-23, IL-12p40, IL-27, IL-5, IL-13, IL-6, IL-9, IL-10, IL-17A, IL-17F, and IL-4) were quantified using LEGENDplexTM Human Th Cytokine Panel (13-plex) (BioLegend 8307643), LEGENDplexTM Human Proinflammatory Chemokine Panel (13-plex) (BioLegend 8314060), LEGENDplexTM Human Cytokine Panel 2 (13-plex) (BioLegend 8302085), and LEGENDplexTM Mouse Th Cytokine Panel

(12-plex) V03 (Biolegend 8315431) following the manufacturer's protocols. Cytokine data acquisition was conducted using a BD FACSCanto II (BD Biosciences, San Jose, CA, USA), and the analysis was performed by LEGENDplex version 8.0 (Biolegend). The serum endotoxin of human and mice was quantitatively assessed by Bioquest Amplite Fluorimetric Endotoxin Detection Kit (60006)

## Metagenomic shotgun sequencing

EP tubes of fresh human feces were maintained at −80℃ for the shotgun metagenomics sequence. Total genome DNA was extracted at Novogene Bioinformatics Technology (Beijing, China) using the sodium dodecyl sulfate (SDS) method. DNA concentration and purity were tested using 1% agarose gels, and sterile water was used to dilute DNA to 1 ng/µL. Briefly, 1% agarose gel was used to monitor DNA degradation and potential contamination. The NanoPhotometer spectrophotometer (Implen, CA, USA) was used for DNA purity detection. The Qubit dsDNA Assay Kit in Qubit 2.0 Fluorometer (Life Technologies, Carlsbad, CA, USA) was used for DNA concentration measurements.

All the human samples were paired-end sequenced on an Illumina platform (insert size 350 bp and read length 151 bp) at Novogene Bioinformatics Technology. As previously described, adapter and low-quality readings were eliminated, and cleaned reads were filtered from human host DNA using the human genome reference (hg19) (27). Afterward, 3,511.15 Gb of high-quality pair-end reads were obtained from 203 human gut microbiota samples at 15.89 Gb per sample.

The high-quality reads were aligned to the updated gut microbiome gene catalog (28) using SOAP2 with a threshold of more than 90% identity over 95% of the length to obtain the functional profile. Sequence-based gene abundance profiling was performed as previously described (29). The relative abundances of Kyoto Encyclopedia of Genes and Genomes (KEGG) orthologous groups were then calculated from their genes. Gene abundance was estimated by counting reads and adjusting by gene length.

## 16S rDNA amplicon sequencing

Total genome DNA from fecal samples of mice was extracted using CTAB/SDS method, and its concentration and purity were evaluated through 1% agarose gels. The universal primers (338F: ACTCCTACGGGAGGCAGCA and 806R: GGACTACHVGGGTWTCTAAT) with the barcode were used for amplifying V3/V4 hypervariable. Specifically, the PCR mixture contained 15 µL of PhusionHigh-Fidelity PCR Master Mix (New England Biolabs), 0.2 µM of forward and reverse primers, and 10 ng of template DNA in a 30 µL reaction volume. Each of the 30 PCR cycles comprised 98℃ for 1 min, 98℃ for 10 s, 50℃ for 30 s, 72℃ for 60 s, and ending 72℃ for 5 min. The PCR products were mixed with the same volume of 1× loading buffer (contained SYB green) and then subjected to quality evaluation using 2.0% agarose gel electrophoresis. The samples with bright main strips (ranging from 400 to 450 bp) were chosen for further analysis. The PCR products were mixed in equidensity ratios and purified using AxyPrepDNA Gel Extraction Kit (Axygen). Sequencing libraries were produced using NEB NextUltraDNA Library Prep Kit for Illumina (NEB, USA) following the manufacturer's recommendations, and index codes were also added. Before the sequencing, the libraries were evaluated for quality using Qubit 2.0 Fluorometer (Thermo Scientific) and Agilent Bioanalyzer 2100 system. The qualified libraries were sequenced on an Illumina Miseq/HiSeq2500 platform, and 250/300-bp paired-end reads were generated.

## Bioinformatics and statistical analysis

Initial data generated by amplicon sequencing must be preprocessed to obtain reliable results for subsequent analysis. The raw data were first subjected to quality evaluation utilizing Trimmomatic software (version 0.33). Subsequently, the Cutadapt software (1.9.1) was used for removing and recognizing primer sequences to acquire clean reads. The clean reads were spliced utilizing Usearch software (version 10), and short sequences (<200 bp) were discarded. Meanwhile, the Uchime (version 4.2) software was used to

identify and delete chimera to obtain the final effective readings. Effective readings were clustered, and Operational Taxonomic Units (OTUs) were subdivided based on a 97% similarity threshold. The rank abundance and rarefaction curves were generated to estimate sequencing depth before a diversities analysis. Gut microbial diversity indices, including Good's coverage, Chao1, ACE, Simpson, and Shannon, were calculated according to each sample's relative abundance of OTU.

Moreover, β diversity analysis was also performed to determine the differences in the main components between groups. Using STAMP analysis, the differential abundance of phyla and genera between B-CS sufferers and healthy populations was studied. Additionally, linear discriminant analysis (LDA) and effect size (LEfSe) was done to determine the Budd-Chiari syndrome-associated taxa in gut microbiota. Statistical analysis of data was conducted using R (version 3.0.3) and GraphPad Prism (version 7.0c). Probability values < 0.05 were considered statistically significant unless otherwise stated.

## RESULTS

### Metagenomics profile of patients with B-CS

To identify whether gut microbial changes are linked to B-CS, we performed shotgun metagenomic sequencing of fecal samples from a cohort of 140 patients and 63 healthy controls. Clinical characteristics of patients with B-CS and healthy controls (Table S1) displayed that the sex ratio, age, and BMI did not differ significantly between the two groups ($P > 0.05$). Consistent with our previous analysis of 16S rDNA from B-CS and healthy controls (10), the community richness at the genus level in the case group was significantly decreased compared to the controls ($P = 0.0058$), and the Shannon index based on the genus profile was calculated to estimate the within-sample diversity of disease groups significantly reduced ($P = 0.00014$) (Fig. 1A). Consistently, the significantly lower community richness ($P = 0.00026$) and α diversity ($P = 0.0017$) at the species level were found in the case group (Fig. 1B). Furthermore, the principal coordinates analysis (PCoA) plot showed that the two groups of gut microbiotas were well distinguished (Fig. 1C). The differences in gut microbiota between case and control groups were like the previous study (10), suggesting a possible deficiency of healthy microflora in B-CS patients.

To analyze the effects of human gut microbiota on B-CS, we compared the gut microbial compositions of both groups at different classification levels. At the phylum level, *Bacteroidetes*, *Proteobacteria*, and *Fusobacteria* in the B-CS group were significantly greater than in the control group ($P < 0.05$) (Fig. 1D). At the genera level, the relative abundances of *Veillonella*, *Prevotella*, and *Campylobacter* in the B-CS patients were observably higher than in the healthy populations ($P < 0.05$). In contrast, the relative abundances of *Bifidobacterium*, *Coprobacillus*, *Enterococcus*, *Lactobacillus*, *Anaerostipes*, and *Lactococcus* were lower ($P < 0.05$) (Fig. 1E). At the species level, the B-CS group exhibited higher abundances of *Veillonella parvula* and *Veillonella atypica* ($P < 0.05$) (Fig. 1F). The difference in gut microbiota composition revealed probable changes in the structure of the intestinal microbiome in both the B-CS and control groups, indicating gut dysbiosis in B-CS patients.

We performed LDA and LEfSe analysis to characterize further which bacterial taxa were distinct between the B-CS and control groups. We identified six genera and seven species showing significant differences (LDA scores > 2.0) (Fig. S1A). A cladogram comparing microbial structure and predominant bacteria between groups was consistent with the above results (Fig. S1B). These data demonstrate significant alterations in the intestinal microbiome structure of the B-CS group, highlighting the significance of gut bacteria in the onset of B-CS.

### Functional alteration in the gut microbiota of B-CS

Using the Kyoto Encyclopedia of Genes and Genomes (KEGG) database (30), we evaluated gut microbial functions across groups in our study cohort. Three hundred

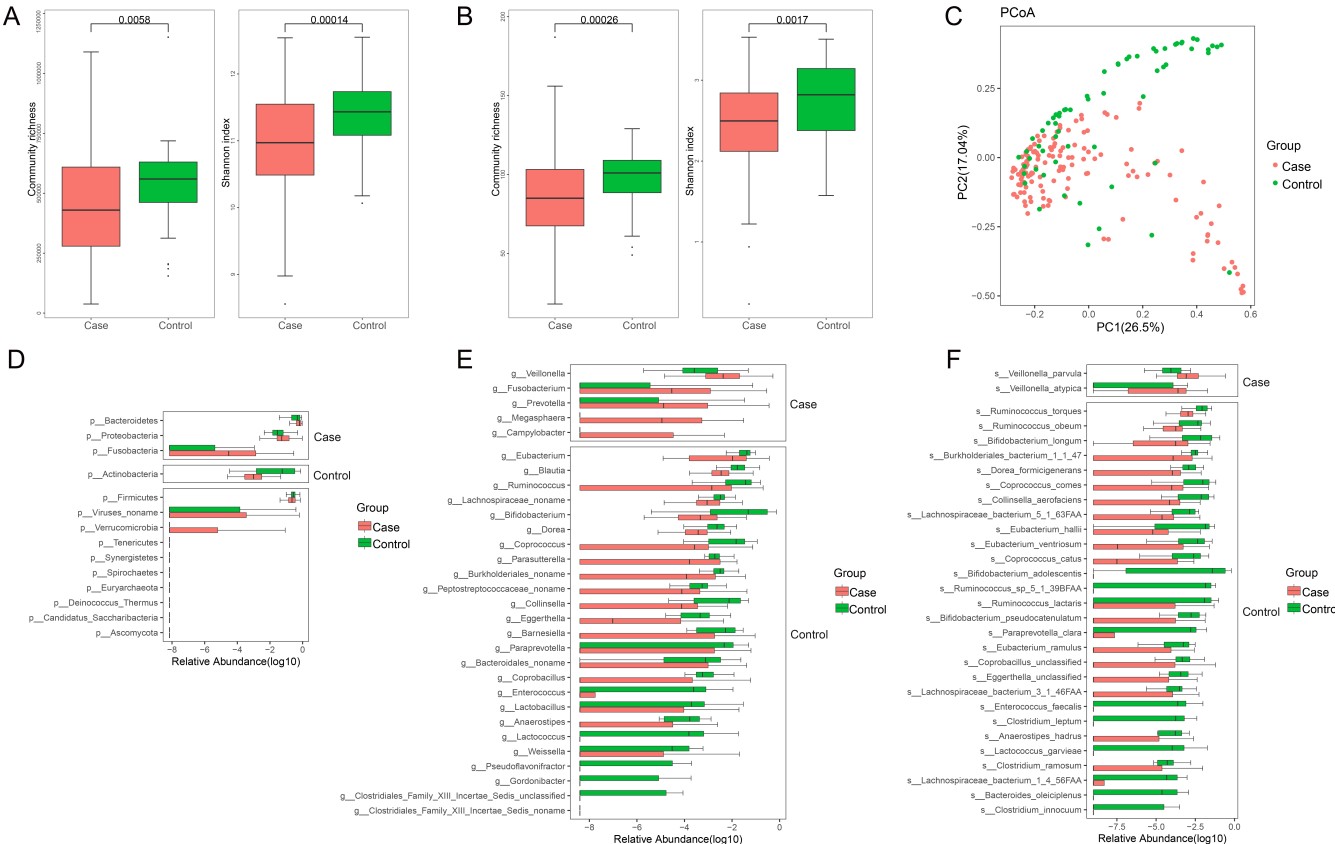

**FIG 1** Dysbiosis of gut microbiota patterns in patients with B-CS and controls. (A) The boxplot shows the community richness and Shannon index at the genus level. (B) The boxplot shows the community richness and Shannon index at the species level. (C) PCoA based on the relative abundance of cases and controls in 203 samples. (D) The boxplot shows the relative abundance of 10 bacteria enriched in the control group at the phylum level. (E) The boxplot shows the relative abundance of five genera enriched in B-CS patients and 26 genera abundant in control. (F) The boxplot shows the relative abundance of two bacteria enriched in B-CS patients and 34 bacteria abundant in control at the species level. Boxes represent the interquartile ranges, and lines inside the boxes denote medians.

twenty-three ($n = 323$) KEGG modules were differentially enriched between the groups ($P < 0.05$) (Table S2). Principal coordinates analysis based on the KEGG orthology revealed notable variations between control and patient microbial activities ($P < 0.05$) (Fig. 2A). Then, gut microbial KEGG pathways compositions were dissected, and microbial functions' shifts in B-CS were identified (Fig. 2B). ABC transporters, lysine biosynthesis, peptidoglycan biosynthesis, leucine and isoleucine biosynthesis, phenylalanine, tyrosine, and tryptophan biosynthesis, histidine metabolism, pyrimidine metabolism, glycine, serine, and threonine metabolism, and cysteine and methionine metabolism were significantly decreased in B-CS groups. These biosynthesis and metabolic functions are essential for the host and have supported the immune system's specific needs (31). Meanwhile, we observed function modules elevated in B-CS, including other glycan degradation, benzoate degradation, bacterial chemotaxis, bacterial secretion system, pentose and glucuronate interconversions, flagellar assembly, and lipopolysaccharide biosynthesis. The gut microbiome's ability to degrade substances and synthesize export LPS has been linked to inflammation and immunological damage (31, 32). Although predictive, functional annotation analysis showed that gut microbiota damage may cause disease by interfering with physiological immune processes.

## Cytokine disorder caused by gut dysbiosis in B-CS

To explore possible immune imbalances in B-CS patients, we collected peripheral blood serum from B-CS patients and healthy controls to detect 39 cytokines. The results indicated that the concentrations of 20 cytokines (IFN-α2, IL-17F, IL-23, IL-27, IP-10, I-TAC,

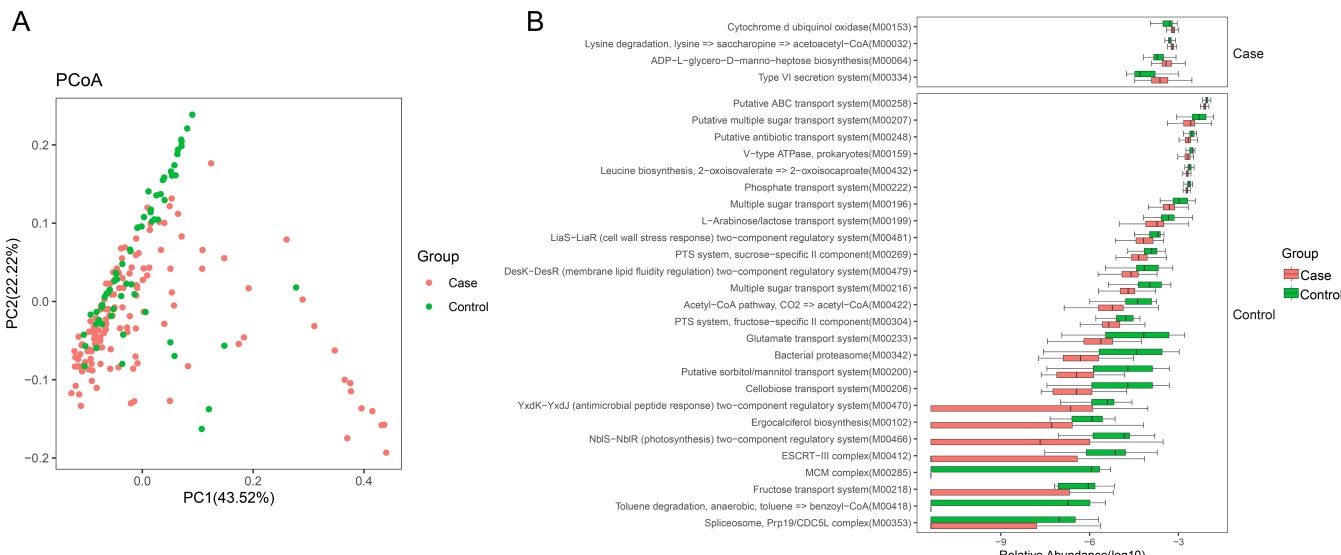

**FIG 2** Functional alteration in gut microbiota of B-CS. (A) Principal coordinate analysis based on unweighted UniFrac distances revealed that the control bacterial communities clustered separately from B-CS bacterial communities. Each point represents a single sample, colored by group. The eigenvalues of axe PCoA1 and PCoA2 were 43.52% and 22.22%, respectively. (B) The boxplot shows differences in gut microbiota annotation function between the B-CS and control groups using the Kyoto Encyclopedia of Genes and Genomes database. Functional pathways related to immunity are enriched in B-CS individuals.

TSLP, MCP-1α, GM-CSF, IL-1α, IL-1β, IL-4, IL-5, IL-6, IL-8, IL-9, IL-10, IL-12p40, IL-13 and IL-17A) in B-CS patients were significantly increased as compared to control populations ($P < 0.001$) (Fig. 3A). B-CS patients' immunological homeostasis is disrupted by abnormal cytokine levels. We then found that B-CS patients had greater serum endotoxin levels than controls (Fig. 3B). We performed an association analysis to investigate further the relationship between gut dysbiosis and cytokine disorder in B-CS patients. At the phylum level, numerous dysregulated cytokines were favorably related to significantly raised microbiota in B-CS and negatively associated with severely reduced microbiota (Fig. 3C).

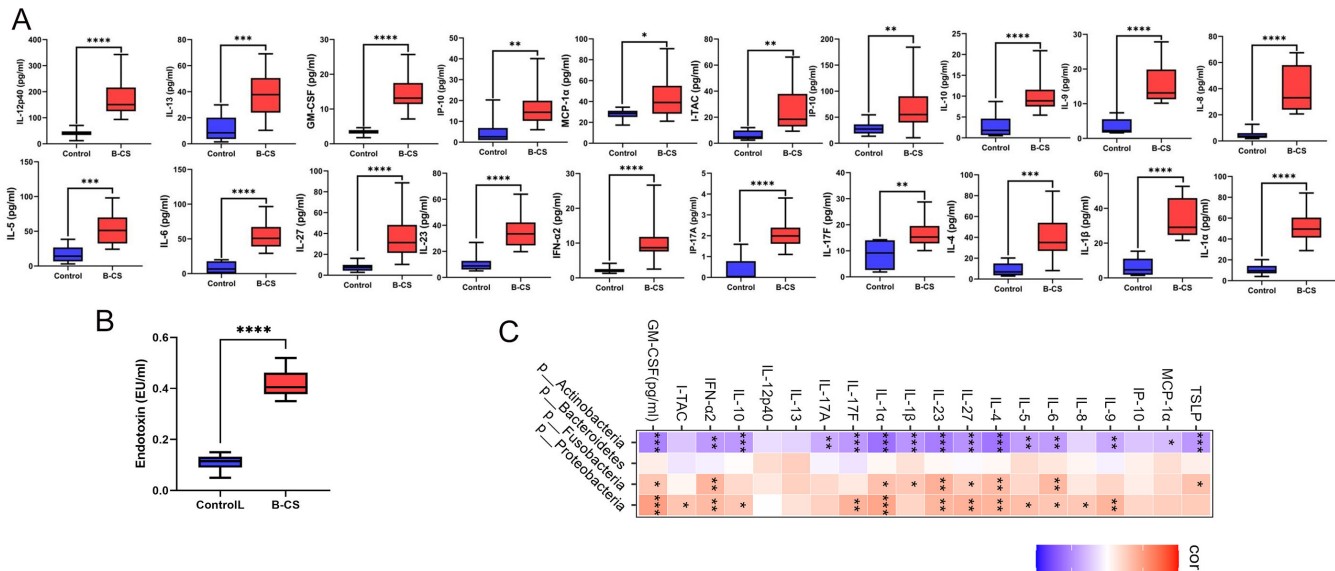

**FIG 3** The association of cytokine disorder and gut dysbiosis in B-CS. (A) B-CS caused significant changes in 20 cytokines (IFN-α2, IL-17F, IL-23, IL-27, IP-10, I-TAC, TSLP, MCP-1α, GM-CSF, IL-1α, IL-1β, IL-4, IL-5, IL-6, IL-8, IL-9, IL-10, IL-12p40, IL-13 and IL-17A) in human peripheral serum. (B) Serum endotoxin levels. (C) The association analysis of cytokine and gut microbiota at the phylum level. *$P < 0.05$, **$P < 0.01$, ***$P < 0.001$.

Consistently, similar results emerged at the genus and species level (Fig. S2A and B). These results reveal that gut dysbiosis in B-CS may cause immune imbalance.

## Gut microbiota composition affected by FMT

Fecal microbiota from B-CS patients were transplanted into mice to study their effects, and the control group received FMT from healthy donors in a similar manner. The FMT procedures were performed once every two days for six weeks (Fig. 4A). The gut microbial profiles of recipient mice were analyzed by 16S rDNA sequencing after transplantation. Twenty fecal samples were collected for amplicon sequencing, and 2240,082 (Con = 1067,384, B-CS = 1172,698) original sequences were acquired. After quality assessment, 1603,547 (Con = 785,516, B-CS = 818,031) high-quality reads were collected, with a median read count of 80,177 (ranging from 60,589 to 94,022) per sample (Table S3). Following taxonomic assignment, the collected sequences were clustered into 1,559 OTUs and 686 OTUs were shared in both groups, accounting for approximately 44.00% of the total OTUs (Fig. S3A). Additionally, 293 and 580 OTUs were uniquely identified in the control and B-CS groups. The tending to smooth species accumulation curves showed that the sample size for this experiment was adequate (Fig. S3B), and the rank abundance curves indicated that the sequencing depth meets the subsequent analysis (Fig. S3C).

Next, sequences were aligned to compute α and β diversities indices. The results of Good's coverage (varying from 99.92 to 99.97%) indicated that most bacterial phenotypes in both groups could be detected. Moreover, there were statistically distinct differences in the ACE ($P = 0.000000317$, $t$-test) and Chao1 ($P = 0.00000082$, $t$-test) indices. In contrast, the Shannon indices ($P = 0.080$, $t$-test) were not significantly different between the Con and B-CS group (Fig. 4 through D). These findings showed that B-CS greatly enhanced mice's gut microbial abundance but did not affect diversity. PCoA plots based on the unweighted UniFrac distances were generated to dissect the variability and similarity of gut microbial main components between both groups, showing a distinct difference (Fig. 4E). Anosim analysis based on the Bray-Curtis algorithm showed that the difference was significantly greater than the within-group difference (R-value = 0.999, $P = 0.001$), validating that the grouping of experiments was meaningful (Fig. 4F). These results showed that despite sharing environment and food, both groups were still segregated, consistent with clustering analyses (Fig. 4G), indicating significant differences in gut microbiota.

The relative abundances of intestinal bacteria at different taxonomical levels were detected, and 10 phyla and 92 genera were recognized in both groups. The *Bacteroidetes* (61.85%), *Firmicutes* (30.92%) and *Proteobacteria* (4.39%) were the three most preponderant phyla in the control group, accounting for over 97% of the taxonomic in the control groups identified (Fig. 4H). Furthermore, *Firmicutes* (57.78%) was the most dominant phylum in the B-CS-FMT groups, followed by the *Bacteroidetes* (38.88%) and *Verrucomicrobia* (1.64%), which together consisted of approximately 98% of the bacterial composition. Other phyla like *Tenericutes* (Con = 0.22%, B-CS = 0.34%), *Cyanobacteria* (Con = 0.41%, B-CS = 0.15), *TM7* (Con = 0.011, B-CS = 0.19) and *Gemmatimonadetes* (Con = 0.0019%, B-CS = 0.00074%) in both groups were found in lower abundances. At the genus level, *Turicibacter* (9.15%), *Lactobacillus* (5.26%) and *Prevotella* (3.94%) were abundantly present in the control group, whereas *Lactobacillus* (32.53%), *Prevotella* (6.91%) and *Oscillospira* (1.68%) were observed as the dominant in the B-CS group (Fig. 4I). In the clustering heatmap, the gut microbial community's distribution and variability were also seen (Fig. S4A and B). These results suggest that the fecal microbiota composition of human donors was successfully reproduced in recipient mice, and the gut microbial abundance and diversity were strongly influenced by the B-CS.

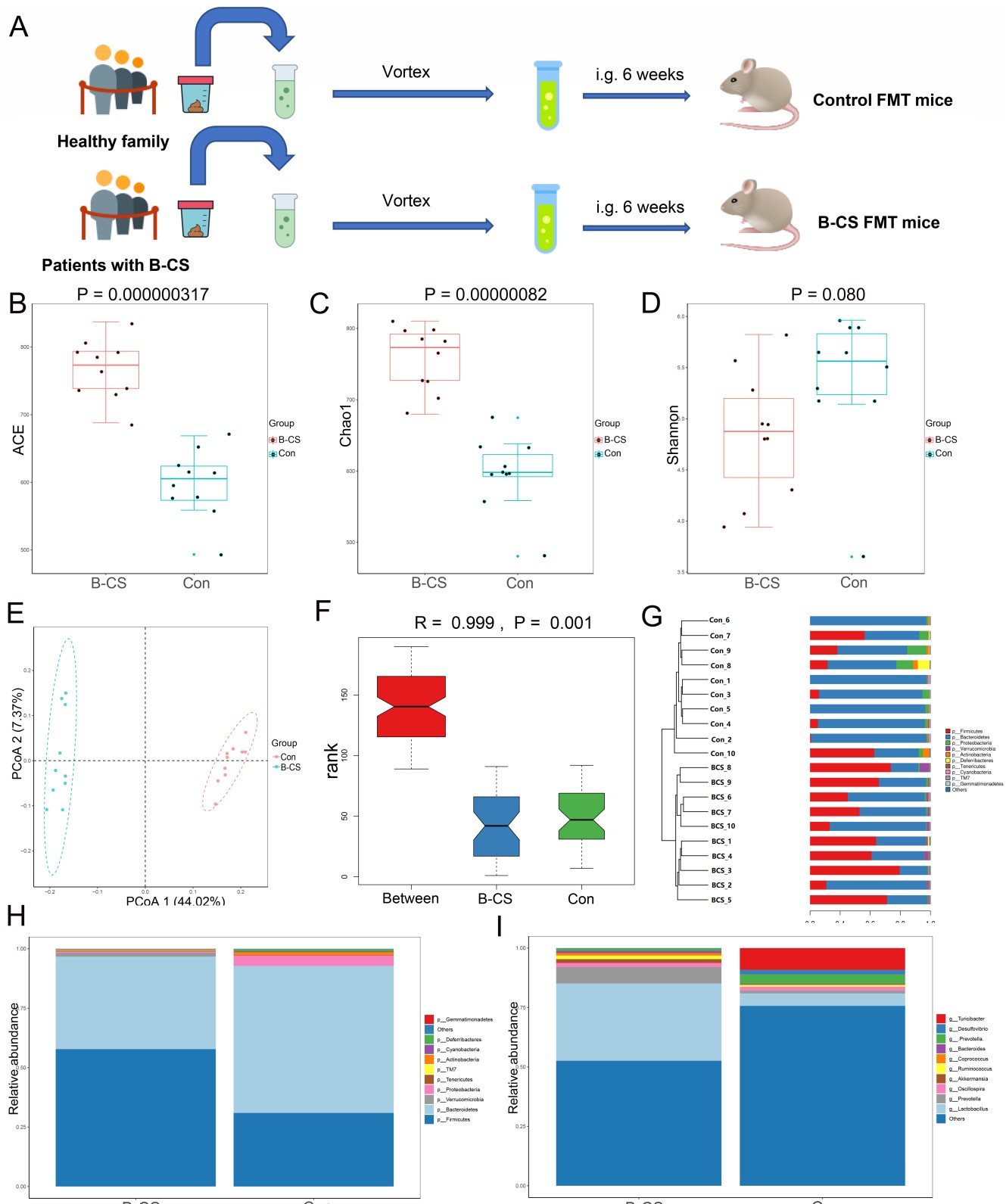

**FIG 4** Administration of fecal bacteria from patients with B-CS to mice induced gut dysbiosis in mice. (A) Schematic representation of fecal microbiota transplantation (FMT), Fresh fecal from three B-CS and three healthy control donors were mixed and used as a single source for B-CS FMT and Control FMT mice, respectively. The recipient mice were randomly divided and orally inoculated daily for three consecutive days and two times per week for 6 weeks with prepared fecal contents. (B–D) ACE Index ($P = 0.000000317$), Chao1 index ($P = 0.00000082$) and Simpson index ($P = 0.080$) in α diversity index analysis (*t*-test). B-CS group

**FIG 4** (Continued)

significantly increased the gut microbial abundance of mice but had no effect on the microbial diversity. (E) Principal coordinate analysis based on unweighted UniFrac distances revealed that the control bacterial communities clustered separately from B-CS bacterial communities. Each circle represents a single sample, coloured by group. The eigenvalues of axe principal co-ordinates analysis (PCoA)one and PCoA2 were 44.02% and 7.373%, respectively. (F) Anosim analysis based on Bray-Curtis algorithm showed that the difference was significantly greater than the within-group difference (R-value = 0.999, P = 0.001). (G) Clustering analysis showed significant differences in the main components of gut microbiota. (H and I) The community structure component diagram demonstrates the community structure of each group at the phylum and genus level. Gut dysbiosis was transferrable by fecal transplant. According to the species annotation results, the top 10 species in the maximum abundance of each grouping were selected to generate a columnar accumulation plot of the relative abundance of species.

## Identification of gut microbial dysbiosis in FMT mice

STAMP analysis (*t*-test) was used for different classification levels to assess gut microbial composition changes in mice after fecal microbiota transplantation. Twenty-seven genera differed between Con and B-CS groups. Of these differential taxa, the relative abundances of 15 genera (*Lactobacillus*, *Bacteroides*, *Christensenella*, *Bilophila*, *Butyricimonas*, *Prevotella*, *Akkermansia*, *rc4-4*, *Tepidibacter*, *Coprococcus*, *Anaerostipes*, *Bacillus*, *Escherichia*, *Anaerofustis* and *Ruminococcus*) memorably increased. In contrast, the relative abundance of 12 bacterial genera (*Odoribacter*, *Sutterella*, *Parabacteroides*, *Alistipes*, *Limnohabitans*, *Hydrogenophaga*, *Corynebacterium*, *Coprobacillus*, *Lampropedia*, *KSA1* and *Staphylococcus*) intensely decreased with the effect of B-CS (*P* < 0.05) (Fig. 5A). Considering this discriminant analysis could not detect all the taxon, LEfSe combined with LDA scores were applied to recognize the specific B-CS-associated bacteria (Fig. 5B and C). Besides the above-mentioned differential bacteria, the richness of *Firmicutes* and *Verrucomicrobia* in the B-CS group was intensely preponderant than control populations. Contrarily, the *Actinobacteria*, *Deferribacteres*, *Proteobacteria* and *Bacteroidetes* were lower. Moreover, we found that some bacterial genera like *Aerococcus*, *Allobaculum*, *Thiothrix*, *Bifidobacterium*, *Mucispirillum*, *Desulfovibrio* and *Turicibacter* were significantly overrepresented in the control group. Contrarily, *Dorea*, *Paraprevotella* and *Clostridium* were preponderant in the B-CS group (*P* < 0.05). LefSe study based on KEGG functional prediction showed significantly different functional pathways between the two groups (Fig. 5D). B-CS FMT group was significantly enriched in Membrane Transport, Carbohydrate Metabolism, Transcription, Xenobiotics Biodegradation and Metabolism, Lipid Metabolism, Immune System Diseases, Metabolism, and Genetic Information Processing. These results were like those in humans, suggesting that more attention needs to be paid to the role of immunity in B-CS.

## Impaired liver and IVC in the B-CS-FMT-induced mice

During the entire FMT experiment, there were no significant likely weight, abdominal circumference, daily urine volume and fecal volume of mice in both groups (Table S4). The liver tissues of control mice were normal according to histopathological results. The B-CS liver tissues showed many hepatocytes ballooning in the central vein, portal area, and liver parenchyma, along with cell swelling and loose, faintly colored vacuolated cytoplasm (Fig. 6A). Moreover, moderate or severe edema and cell swelling were found in the medial layer of the posterior hepatic vein vessels in the mice belonging to the B-CS-FMT group as compared to controls (Fig. 6A). Meanwhile, compared to normal controls, B-CS patients had higher blood endotoxin levels. (Fig. 6B). To dissect the changes in the tissue, we observed the ICAM-1, vWF, CD31 and CD34 expression under a fluorescence microscope to reflect the changes in vascular tissue (27, 28, 33). Immuno-fluorescence results indicated that the fluorescence intensity of ICAM-1, vWF, CD31 and CD34 in the IVC of B-CS-FMT mice was higher than the controls (Fig. 6C). Additionally, the fluorescence intensity of ICAM-1 and vWF in the liver of B-CS mice was higher than the controls, whereas CD31 and CD34 were lower (Fig. 6D). These results show that B-CS-like phenotypes were induced by fecal bacterial transfer from B-CS patients.

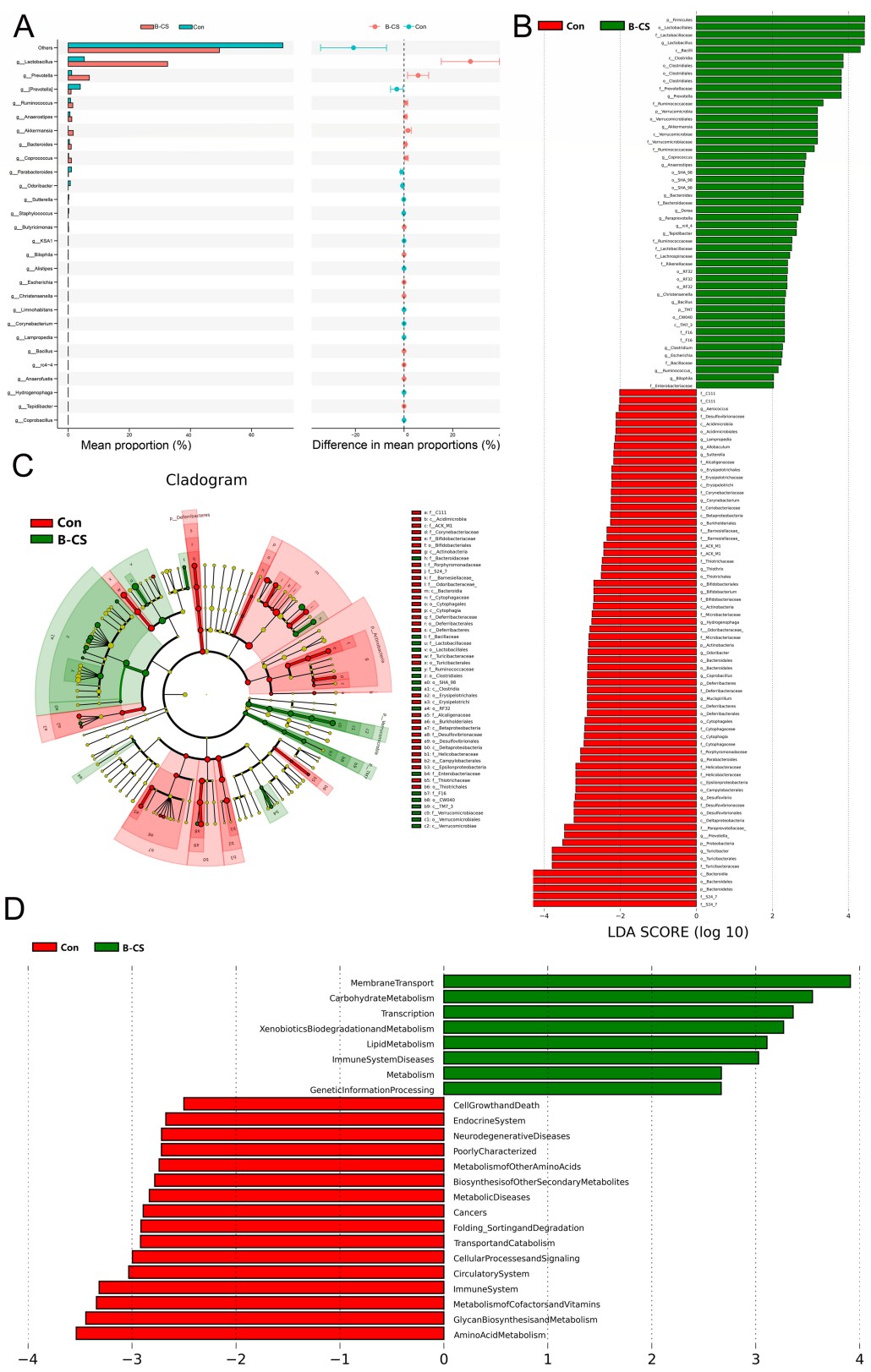

**FIG 5** Differences in gut microbiota between control and B-CS-FMT groups. (A) Analysis of STAMP differences (*t*-test) at the genus level showed the abundance of species between the two groups. The left panel showed the proportion of abundance of different species in two samples or two groups of samples, the middle showed the proportion of difference within 95% CI, the rightmost value is *P*-value, and *P* value < 0.05 indicates significant difference. (B) linear discriminant

**FIG 5** (Continued)

analysis (LDA) effect size identified the most differentially abundant taxa between the two groups. Only taxa meeting an LDA significant threshold of >2 are shown. (C) Differential species evolutionary cladogram recognized the specific bacteria associated with B-CS. (D) The results of LefSe analysis based on KEGG functional prediction, including histograms of LDA value distribution and functional items with statistical differences in different abundance comparison plots.

## B-CS-FMT impaired the intestinal barrier and imbalanced the cytokines of mice

It comes to light that disturbed gut microbiota could impair the intestinal barrier and enhance intestinal permeability (34). A histological examination was performed on colonic tissues. H&E staining displayed lymphocytic infiltration in the colonic mucosal layer, mild edema and widened inter-glandular distance of the B-CS group (Fig. 7A). These findings suggested a colonic barrier breach. Immunohistochemistry and immunofluorescence studies showed B-CS-FMT colon ZO-1 and Occludin levels were lower than the control group (Fig. 7B and C), indicative of an impaired intestinal barrier condition. Taken together, these data demonstrate that the intestinal barrier may be disrupted by fecal bacteria in B-CS patients.

Our above work showed that B-CS-like phenotypes and intestinal barrier impairment were found in mice after FMT and immune-linked microbial functions were enriched in B-CS patients and B-CS-FMT mice. Then, we collected serum samples of mice for cytokine detection. Analogously, the B-CS-FMT mice possessed higher levels of IL-5, IL-6, IL-9, IL-10, IL-17A, IL-17F and IL-13 compared to healthy controls ($P < 0.05$ or $P < 0.01$) (Fig. 7D). These significantly increased cytokines are consistent with human variances. These data showed that B-CS patients' fecal microbiota alter the immune response, which causes B-CS development.

## DISCUSSION

B-CS is a rare disease that may induce liver failure or death (35). However, its etiology is unknown, and no reliable procedures exist to forecast, prevent, or treat it (1). It has been proven that gut microbiota plays essential roles in host health, immunity, and metabolism, and this topic has garnered growing interest (12). Moreover, some recent studies have also revealed the key roles of gut microbiota in liver diseases by mediating the gut-liver axis (36, 37). However, its potential influence on B-CS remains uncertain. To make up for these blanks, we applied a strategy based on metagenomic analyses, coupled with FMT, to investigate the potential relationship between the gut microbiota and the pathogenesis of B-CS in this study, and these results established that the immune imbalance caused by gut dysbiosis was important driving factors for B-CS development.

In the present study, gut dysbiosis in B-CS observed and echoed the findings of our previous reports (10). Our findings show that B-CS was linked to decreased bacterial diversity and variation. In this study on the gut microbiota of B-CS, we combined metagenomic sequencing of re-collected samples with previous 16S sequencing results to study microbial community composition structure, diversity, and function more efficiently and accurately. Functional links have been well investigated and showed that gut dysbiosis in B-CS were associated with immune homeostasis imbalance. We also found significantly altered α and β diversity in the gut microbiota of the B-CS-FMT mice compared to the control mice, and the control mice received FMT from healthy donors. Our gut microbial diversity research showed that the B-CS-FMT group differed from the controls in the main gut microbiota components despite a shared diet and environment. Consequently, we suggested that fecal transplant was the major driving force for gut microbial dysbiosis of mice, and the dysbiosis pattern in B-CS reported in our research is reliable. Our latest evidence, including metagenomics analysis of B-CS populations and FMT animal model inquiry results, suggested that gut dysbiosis were important factors contributing to the development of B-CS. Therefore, gut microbiota must be considered when treating B-CS.

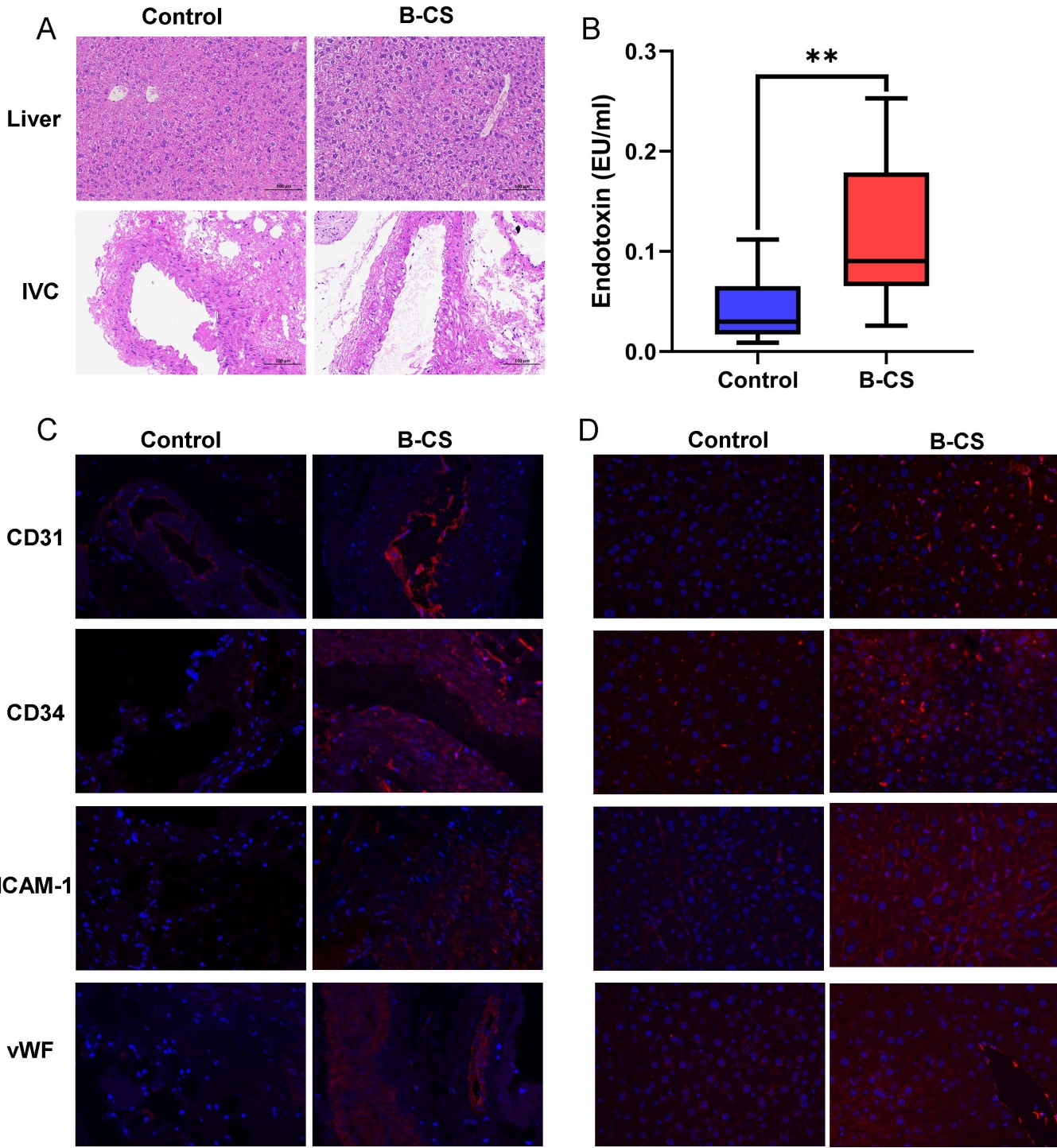

FIG 6 Fecal microbiota transplanted from the patients with B-CS affected hepatic and IVC function in recipient mice. (A) Representative H&E staining of the liver and IVC of mice. Original magnification, ×200. (B) Serum endotoxin levels. *$P < 0.05$, **$P < 0.01$ (C) Representative image of CD31, CD34, ICAM-1 and vWF immunofluorescence staining in IVC tissues of post-hepatic segment (original magnification, ×400). (D) Representative image of CD31, CD34, ICAM-1 and vWF immunofluorescence staining in liver tissues (original magnification, ×400).

Gut dysbiosis could selectively stimulate the proliferation of pathogenic bacteria, which disturb host immunity and intestinal barrier function (38) and induce multiple diseases such as diabetes, metabolic syndrome and non-alcoholic fatty liver (20, 21). Gut dysbiosis resulting from B-CS-FMT has been found to cause a notable rise in

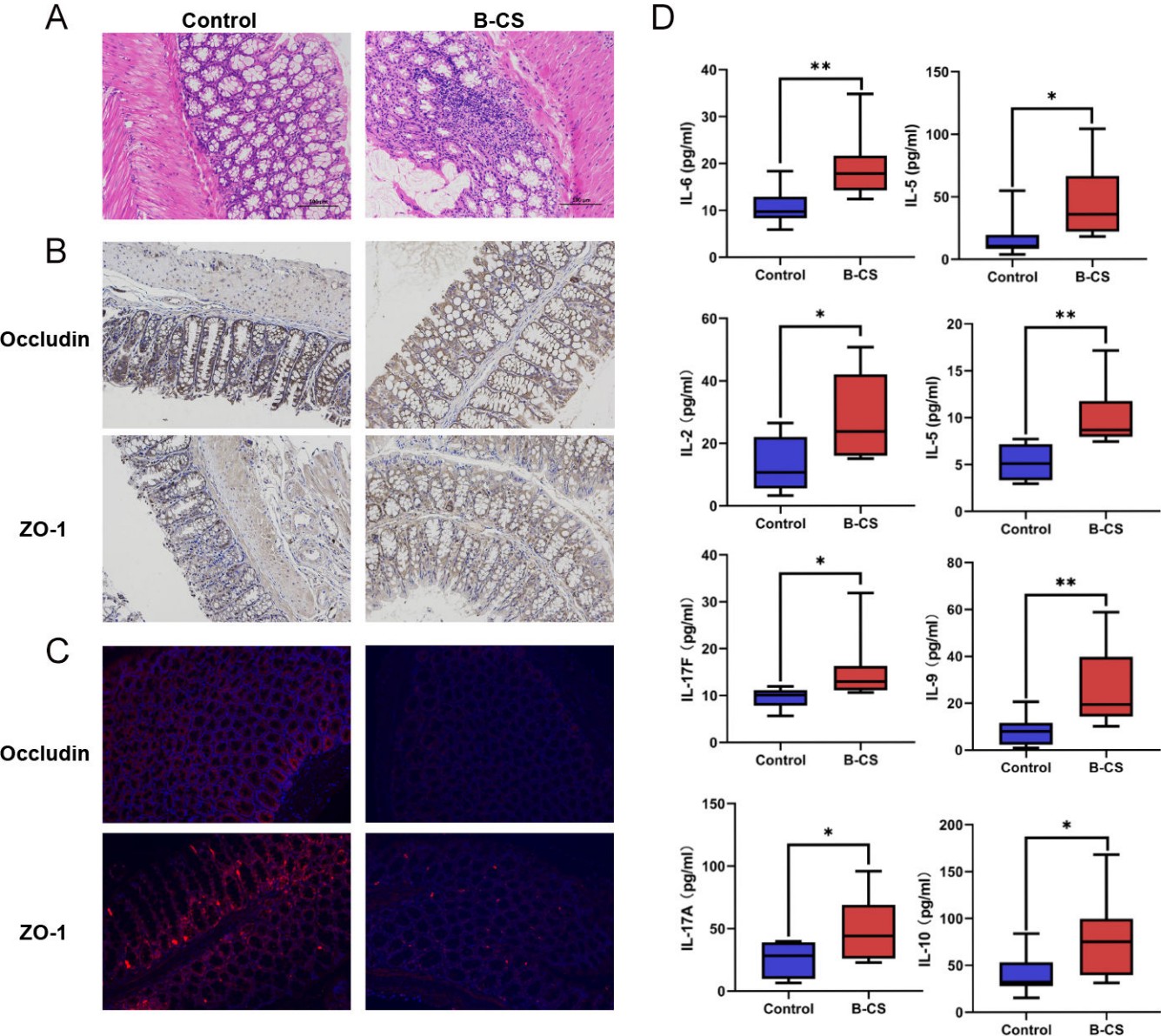

**FIG 7** CS-FMT impaired intestinal barrier and imbalanced cytokines of mice. (A) Histological examination of colon. The sections were stained with H&E. Original magnification, ×200. (B) Immunohistochemistry determination of ZO-1 and Occludin expression in the colon tissue of mice. Original magnification, ×200. (C) Immunofluorescent analysis of ZO-1 and Occludin in the colon tissue of mice. Original magnification, ×200. (D) A total of 7 cytokines (IL-5, IL-6, IL-9, IL-10, IL-17A, IL-17F and IL-13) were found to be significantly altered in the peripheral serum of B-CS-FMT mice. *$P < 0.05$, **$P < 0.01$, ***$P < 0.001$.

potentially pathogenic bacteria, which play a crucial role in the development of B-CS. Moreover, for some bacterial taxa containing opportunistic pathogens, we also found that the relative abundances of *Prevotella*, *Veillonella* and *Campylobacter* in B-CS patients increased significantly. In contrast, *Prevotella* and *Bacteroides* in B-CS-FMT mice were increased intensely. *Veillonella* was altered in its associations with the immunity and metabolism of patients with primary biliary cirrhosis (39). The enrichment of *Campylobacter*, remarkably *Campylobacter concisus*, in the intestines of B-CS patients has caused a substantial increase in the incidence and prevalence of campylobacteriosis in developed and developing countries (40). Previous research showed that *Campylobacter* may connect and penetrate human intestinal epithelial cells and macrophages, compromise intestinal barrier integrity, release toxins, and strategically elude host immune responses (41). *Clostridium* and *Bacteroides* have been demonstrated to be closely associated with

gut microbial dysbiosis or infection (42, 43). This study showed that gut microbiota composition is altered and that these microbes may cause B-CS.

Functional enrichment and correlation analyses showed that gut dysbiosis is linked to an immunological imbalance in B-CS populations and B-CS-FMT mice. This study found several human and mouse microorganisms different from controls. *Prevotella* could translocate to the liver and cause autoimmunity in genetically predisposed hosts (44). Along with alterations in gut microbial composition, we observed dysregulation of bacterial gene function. The gut microbiota of B-CS patients and B-CS-FMT mice were depleted in genes associated with the biosynthesis and metabolism of amino acids, such as lysine, histidine, leucine, and serine, which are essential for health. Contrarily, the abundance of endotoxin biosynthesis and export modules in B-CS patients and B-CS-FMT mice is potentially important for gut microbiota in causing low-grade inflammation. Inflammation caused by endotoxin-triggered immune responses is an essential feature of Gram-negative bacteria pathogenesis, like *Prevotella* (45, 46). Consistently, *Prevotella* has shown the same trend in the B-CS population and mice in this study, and emerging studies in humans and mice have linked the increased abundance of *Prevotella* to localized and systemic disease, as demonstrated by augmented release of inflammatory mediators from immune cells and various stromal cells (47, 48). Thus, *Prevotella* may be vital in B-CS by triggering immune responses.

As known, gut microbiota dysbiosis can elicit host immune response, including chronic inflammation (22). This investigation found inflammatory symptoms in B-CS-FMT mice's liver and adjacent vessels, comparable with histological findings in B-CS patients (49). Moreover, we also found the infiltration and aggregation of lymphocytes in the colonic mucosa, indicating local inflammation in the colon. The increased intestinal permeability and impaired intestinal barrier were confirmed in B-CS-FMT mice, and systemic inflammation in B-CS individuals and B-CS-FMT mice was exhibited, which inferred the existence of immune imbalance. The breakdown of the intestinal barrier had enabled pathogens and toxins to infiltrate tissues, organs, and the circulatory system (34) . This infiltration had been associated with increased inflammatory cytokines, immune-modulation, and altered protein expression, which could lead to hepatic injury, vascular dysfunction, and portal hypertension (50, 51). Pathogenic bacteria and their endotoxin could cause immunological imbalance, strongly associated with gut microbial dysbiosis in B-CS patients and B-CS-FMT mice, which produced or aggravated B-CS by crossing the defective intestinal barrier. This study first revealed cytokine imbalance, liver and IVC alterations, and intestinal barrier impairment in B-CS-FMT mice and provided a novel insight of mechanism for elucidating the B-CS pathogenesis.

This study has revealed that gut dysbiosis is a potential contributing factor in the development of B-CS. The dysbiosis disrupted intestinal permeability, triggering immune system activation through the production of toxic metabolites and imbalanced cytokines. Consequently, the escalation of causative factors led to their concentration in the portal vein, thereby compromising both the liver parenchyma and outflow tract. Therefore, we propose that gut dysbiosis induces immune imbalance, leading to chronic systemic inflammation, which further facilitates the progression of B-CS. Moreover, our findings suggested that *Prevotella*, a type of gut bacteria, may play a crucial role in promoting inflammation and immune imbalance, thereby implicating it in the pathogenesis of B-CS. It is important to note that this study is the first of its kind to establish a connection between B-CS and gut dysbiosis. However, due to the complex interplay between the host and gut microbiota, further research is warranted to comprehensively understand the potential role of gut microbiota in B-CS.

## ACKNOWLEDGMENTS

All human samples were obtained from the Biobank of the First Affiliated Hospital of Zhengzhou University and National Human Genetic Resources Sharing Service Platform (Grant No. 2005DKA21300). We thank Dr. Hongyan Ren of Shanghai Mobio Biomedical Technology Co., Ltd for her discussion on the writing of the paper.

The project was supported by the National Natural Science Foundation of China (Nos. 81870457, 82172944 and 81900558) and Key science and technology projects of Henan Province (No. 232102311048).

All persons who meet authorship criteria are listed as authors. #These authors contributed to the work equally and should be regarded as co-first authors. The project was designed by Y.S. The project was managed by R.Z., W.W., R.L. and Q.L. Q.L., L.Z., R.Z., Z.L., P.X., T.B., Z.W., G.W., J.R., D.J., Y.S., X.M., J.Z., K.D., L.W., D.Z., G.J. and H.Z. contributed to acquisition of clinical samples, patients' information and clinical data analyses. Y.S., L.Z., R.Z. and Q.L. performed the data analysis. Y.S. and Q.L. wrote the paper. All authors read and approved the final manuscript.

## AUTHOR AFFILIATIONS

[1]Department of Hepatobiliary and Pancreatic Surgery, the First Affiliated Hospital of Zhengzhou University, Zhengzhou, China

[2]Institute of Hepatobiliary and Pancreatic Diseases, Zhengzhou University, Zhengzhou, China

[3]Key Lab of Hepatobiliary and Pancreatic Diseases, Zhengzhou, China

[4]Department of Gastroenterology, the First Affiliated Hospital of Zhengzhou University, Zhengzhou, China

[5]Department of Ultrasound Diagnosis, the First Affiliated Hospital of Zhengzhou University, Zhengzhou, China

[6]Department of Endovascular Surgery, the First Affiliated Hospital of Zhengzhou University, Zhengzhou, China

[7]Department of Interventional Radiology, the First Affiliated Hospital of Zhengzhou University, Zhengzhou, China

[8]Department of Vascular Surgery, the First Affiliated Hospital of Zhengzhou University, Zhengzhou, China

## AUTHOR ORCIDs

Yuling Sun http://orcid.org/0000-0001-5289-4673

## FUNDING

| Funder | Grant(s) | Author(s) |
|---|---|---|
| MOST | National Natural Science Foundation of China (NSFC) | 81870457 | Yuling Sun |
| MOST | National Natural Science Foundation of China (NSFC) | 82172944 | Yuling Sun |
| MOST | National Natural Science Foundation of China (NSFC) | 81900558 | Weijie Wang |
| Key science and technology project s of Henan Province | 232102311048 | Ruopeng Liang |

## AUTHOR CONTRIBUTIONS

Qinwei Lu, Investigation, Methodology, Writing – original draft, Writing – review and editing | Rongtao Zhu, Investigation, Methodology, Writing – review and editing | Lin Zhou, Resources, Supervision | Ruifang Zhang, Resources, Supervision | Zhen Li, Resources, Software | Peng Xu, Resources, Supervision | Zhiwei Wang, Resources, Supervision | Gang Wu, Resources, Supervision | Jianzhuang Ren, Resources, Supervision | Dechao Jiao, Resources, Supervision | Yan Song, Resources, Supervision | Jian Li, Resources, Validation | Weijie Wang, Resources, Validation | Ruopeng Liang, Resources, Validation | Xiuxian Ma, Resources, Validation | Yuling Sun, Conceptualization, Funding acquisition, Writing – original draft, Writing – review and editing

## DATA AVAILABILITY

All data from this study are publicly available. The metagenomics data of human are available at GSA (https://ngdc.cncb.ac.cn/gsa-human/) where it has been assigned the ID (HRA005588). The 16S rDNA sequencing data of mice are available at NCBI, where it has been assigned the ID (PRJNA786092).

## ETHICS APPROVAL

The study protocol was approved by the Ethics Committee of the First Affiliated Hospital of Zhengzhou University (2021-KY-0596), and all patients signed an informed consent form before their specimens were selected according to the rules of the Ethics Committee.

## ADDITIONAL FILES

The following material is available online.

### Supplemental Material

**Supplemental figures (mSystems00794-24-s0001.pdf).** Figures S1 to S4.
**Table S1 (mSystems00794-24-s0002.xlsx).** Clinical characteristics.
**Table S2 (mSystems00794-24-s0003.xlsx).** KEGG modules differentially enriched between groups.
**Table S3 (mSystems00794-24-s0004.xlsx).** Quality assessment data of fecal samples.
**Table S4 (mSystems00794-24-s0005.xlsx).** Basic mouse data.

### Open Peer Review

**PEER REVIEW HISTORY (review-history.pdf).** An accounting of the reviewer comments and feedback.

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
