## [Reviewer comments · mSystems]

Gut dysbiosis contributes to the development of Budd-Chiari syndrome through immune imbalance

Qinwei Lu, Rongtao Zhu, Lin Zhou, Ruifang Zhang, Zhen Li, Peng Xu, Zhiwei Wang, Gang Wu, Jianzhuang Ren, Dechao Jiao, Yan Song, Jian Li, Weijie Wang, Ruopeng Liang, Xiuxian Ma, and Yuling Sun

Corresponding Author(s): Yuling Sun, The First Affiliated Hospital of Zhengzhou University

Review Timeline:

Submission Date:	June 11, 2024
Editorial Decision:	July 1, 2024
Revision Received:	July 4, 2024
Accepted:	July 17, 2024

Editor: Katrine Whiteson

Reviewer(s): The reviewers have opted to remain anonymous.

Transaction Report:

DOI: <https://doi.org/10.1128/msystems.00794-24>

Re: mSystems00794-24 (Gut dysbiosis contributes to the development of Budd-Chiari syndrome through immune imbalance)

Dear Prof. Yuling Sun:

Thank you for the privilege of reviewing your work. We appreciate your attention to the reviewer comments.

Following Editorial review, we request a minor revision which affects several of the figures - to please increase the font size of figure legends and text in the figures to increase readability.

Below you will find my comments, instructions from the mSystems editorial office, and the reviewer comments.

Figure legend text size should be increased to improve readability

Revision Guidelines

Sincerely,
Katrine Whiteson
Editor
mSystems

Dear Editor and Reviewer:

Thank you for allowing us to submit a revised draft of my manuscript titled “**Gut dysbiosis contributes to the development of Budd-Chiari syndrome through immune imbalance**” to *mSystems* [mSystems00794-24]. We appreciate the time and effort that you and the reviewers have dedicated to providing your valuable feedback on my manuscript. We are grateful to the reviewers for their insightful comments on my paper. We have been able to incorporate changes to reflect most of the suggestions provided by the reviewers. We have highlighted the changes within the manuscript.

Here is a point-by-point response to the editorial office and the reviewer's comments.

Comment: Figure legend text size should be increased to improve readability.

Response: Thank you for your valuable feedback on our manuscript. We increased the font size of the text in the legend and figure comprehensively to increase readability (Figure 1-7 and Supplementary figures).

We look forward to hearing from you in due time regarding our submission and to respond to any further questions and comments you may have.

Thank you.

Sincerely,

Yuling Sun (Ph.D., M.D.)

Department of Hepatobiliary and Pancreatic Surgery, First Affiliated Hospital, Zhengzhou University, School of Medicine, 1 Jianshe Road, Zhengzhou, China

Email : ylsun@zzu.edu.cn

Re: mSystems00794-24R1 (Gut dysbiosis contributes to the development of Budd-Chiari syndrome through immune imbalance)

Dear Prof. Yuling Sun:

Your manuscript has been accepted, and I am forwarding it to the ASM production staff for publication. Your paper will first be checked to make sure all elements meet the technical requirements. ASM staff will contact you if anything needs to be revised before copyediting and production can begin. Otherwise, you will be notified when your proofs are ready to be viewed.

Sincerely,
Katrine Whiteson
Editor
mSystems